# Clinical Applications, Pitfalls, and Uncertainties of Thrombin Generation in the Presence of Platelets

**DOI:** 10.3390/jcm9010092

**Published:** 2019-12-30

**Authors:** Marina Panova-Noeva, Paola E.J. van der Meijden, Hugo ten Cate

**Affiliations:** 1Clinical Epidemiology and Systems Medicine, Center for Thrombosis and Hemostasis (CTH), University Medical Center of the Johannes Gutenberg-University Mainz, 55131 Mainz, Germany; Marina.Panova-Noeva@unimedizin-mainz.de; 2Department of Biochemistry, Cardiovascular Research Institute Maastricht (CARIM), Maastricht University Medical Center, 6200 Maastricht, The Netherlands; p.vandermeijden@maastrichtuniversity.nl; 3Department of Internal Medicine, Laboratory of Clinical Thrombosis and Haemostasis, and Cardiovascular Research Institute Maastricht, Maastricht University Medical Center, 6200 Maastricht, The Netherlands

**Keywords:** platelets, thrombin generation, cardiovascular risk factors, cardiovascular diseases, thrombosis, bleeding, hemophilia, von Willebrand disease

## Abstract

Platelet-dependent thrombin generation is a helpful tool to assess ex vivo the interaction between platelets and plasma coagulation factors in the initiation, amplification, and inhibition of thrombin generation (TG). This review article discusses the most relevant available data on the clinical applications of fluorogenic TG, the most widely used TG assay, performed in the presence of platelets, i.e., in platelet-rich plasma. With respect to prothrombotic states, arterial hypertension and obesity were the most prominent cardiovascular conditions linked to increased platelet-dependent TG. In addition, platelet-associated hypercoagulability, assessed by the TG assay, has been shown in individuals with active cancer. In terms of bleeding, platelet-dependent TG has been applied to assess bleeding risk in individuals with hemophilia, von Willebrand disease, and Glanzmann thrombasthenia as well as in subjects with other congenital or acquired coagulation factor deficiencies. In addition to risk prediction, a role of the TG assay has been suggested in monitoring antiplatelet therapy in prothrombotic conditions and replacement therapy in bleeding diathesis. Finally, for the routine clinical use and as a biomarker of disease development and progression, better standardization and clinical validation of platelet-dependent TG are still needed.

## 1. Introduction

Thromboembolic disease and bleeding, representing two sides of the same coin, have been traditionally investigated by using different laboratory approaches. In addition to the clinical probability, D-dimers are currently the only established laboratory biomarkers routinely used to rule out venous thromboembolism (VTE) [1]. On the other hand, activated partial thromboplastin time (aPTT), prothrombin time (PT), and platelet function investigations are in clinical use for evaluating bleeding diathesis [2]. The thrombin generation (TG) assay has emerged as a global coagulation tool that can assess both hypercoagulable and hypocoagulable conditions [3]. It can be performed in the absence of platelets, i.e., platelet-poor (PPP) and platelet-free plasma (PFP), and in the presence of platelets, i.e., platelet-rich plasma (PRP). TG in PRP enables the investigation of interactions between platelets and coagulation factors in plasma, thus representing an assay that closely mimics in vivo conditions [4]. The calibrated automated thrombogram (Thrombinoscope BV, Maastricht, The Netherlands), the most frequently used TG assay, employs a standardized testing procedure where readings from a fluorometer are recorded and calculated by means of the Thrombinoscope software (Thrombinoscope BV). In principle, the measurements in PRP are assessed in fresh material, whereas TG in PFP is assessed in frozen, stored material. For TG measurements in PRP, coagulation is triggered by adding 20 μL exogenous tissue factor (TF) to 80 μL PRP (with adjusted platelet concentration of 150,000 platelets/μL), whereas for TG measurements in PFP, 20 μL exogenous TF and 4 μM phospholipids are added to 80 μL plasma. After prewarming for 10 min at 37 °C in the fluorimeter, the reaction is started by adding 20 μL of a low-affinity fluorogenic substrate for thrombin (Z-Gly-Gly-Arg-AMC) and calcium chloride mixture (FluCa). TG is measured as a function of a calibration signal obtained in a sample from the same plasma (PRP or PFP) after addition of a fixed amount of thrombin–α2-macroglobulin complex (20 μL of thrombin calibrator) and 20 μL of FluCa. The TG method is extensively reviewed in [5,6,7]. Parameters derived from a TG curve and most frequently reported in the studies are lag time, time to minimum thrombin formed expressed in minutes (min); peak height, maximum concentration of thrombin formed expressed in nM thrombin; and endogenous thrombin potential (ETP) or area under the curve expressed as nM of thrombin formed per minute.

The aim of this article was to review the available evidence for the potential clinical application of platelet-dependent TG in both prothrombotic and bleeding conditions, as depicted in Figure 1. Aspects preventing its broader application and routine clinical practice will also be discussed as uncertainties and limitations of thrombin generation in the presence of platelets.

## 2. Platelet-Dependent Thrombin Generation in Prothrombotic Conditions

The evidence for the application of platelet-dependent TG in a clinical prothrombotic setting is rather limited. Platelet-dependent TG has been a subject for investigation in individuals presenting with most typical clinical prothrombotic conditions, such as traditional cardiovascular risk factors (CVRFs) and cardiovascular diseases (CVD).

The link between platelet-dependent TG and arterial hypertension has been recently addressed in an experimental mouse model with angiotensin II-induced hypertension, demonstrating a role for glycoprotein Ibα on platelets in factor XI (FXI)-dependent amplification of TG [8]. The evidence for increased platelet-dependent TG in hypertensive individuals has been less straightforward. In a small sample of hypertensive individuals (*n* = 71), a difference in ETP in PRP was observed between subjects with controlled arterial hypertension (*n* = 17) and stage II or higher arterial hypertension (*n* = 35) with limited statistical power [8]. Results from the large population-based Gutenberg Health Study (GHS), including 407 individuals, showed a positive association between TG peak height in the presence of platelets and newly diagnosed arterial hypertension. In the multivariable analysis, accounting for all traditional CVRFs, history of CVD, age, and sex, no independent association was observed between TG and arterial hypertension [9]. In a study of childhood cancer survivors, arterial hypertension was identified as an important determinant of higher platelet-dependent TG, independent of the known potential confounders [10]. No associations were observed for arterial hypertension and TG in the absence of platelets. The possible reason for the lack of association in individuals from the GHS might be result of the intake of antihypertensive medications, unlike childhood cancer survivors that were mainly young with unrecognized and untreated arterial hypertension [11]. Therefore, the evidence available so far, supports platelet-dependent hypercoagulability predominantly in subjects with uncontrolled and/or untreated arterial hypertension. Furthermore, the results suggest a pleotropic role for antihypertensive medications beyond lowering the blood pressure, with important effects on interactions between platelet and coagulation factors.

Obesity, presently the largest global health problem in adults, has been associated with a hypercoagulable state resulting from increased platelet activation and thrombin generation, and decreased fibrinolysis [12,13,14]. The results on TG from the population-based GHS identified obesity as the most important determinant of higher ETP in both PRP and PFP, independent of the traditional cardiovascular risk factors with greater strength of evidence for the association in the presence of platelets [9]. Studies of TG in PPP have confirmed obesity-related hypercoagulability that could also be partly explained by residual platelets and/or platelet-derived microparticles in plasma. Indeed, it has been demonstrated that increased TG in PPP of obese individuals correlate with platelet-derived microparticles and has a linear correlation with body mass index (BMI) and waist circumference [15]. Correction of BMI has been linked with considerable reduction in TG profile, assessed in one-time centrifuged PPP [16]. How much of the reduction in TG can be explained by changes in platelet activation status is currently unknown. Interestingly, whereas obesity consistently showed an association with TG, it failed to associate with other global markers of in vivo hypercoagulability, such as prothrombin fragment 1, fibrinopeptide A, and thrombin–antithrombin complex (TAT) [17]. This further potentiates the role of platelet-derived factors and their contribution to TG potential in subjects with obesity.

Platelet activation has been recognized in subjects with ischemic stroke including coronary artery disease, and is regarded as a key player in the pathogenesis of atherosclerosis development and progression [18]. Studies investigating platelet-dependent TG in these populations are limited and the lack of platelets in the investigated sample could explain the weak associations between TG in PFP and cardiovascular outcomes [19]. It has been reported, by applying an older subsampling technique, that TG investigated in the PRP of young stroke patients was higher compared to control subjects. The same comparisons in PPP showed no differences in TG potential between young stroke patients and controls [20]. Within patients with an acute transient ischemic attack, TG was compared between subjects with intracranial atherosclerotic disease (ICAD), one of the leading causes of ischemic stroke, and those without ICAD. Interestingly, no differences in TG were observed both in the presence and absence of platelets between subjects with and without ICAD. Only after the addition of thrombomodulin, a hypercoagulable state could be detected in the ICAD group [21].

It has been reported that the TG assay is a suitable tool to assess venous thromboembolism (VTE)-related hypercoagulability, both in PPP and PRP [22]. Platelets have been implicated in the pathogenesis of VTE; however, data on platelet-dependent TG in subjects at risk for VTE are lacking. Plasma hypercoagulability assessed by the TG assay in the absence of platelets has been a subject of intensive investigation in individuals at risk for first unprovoked and recurrent VTE [23,24,25]. In a rather small group of individuals with a history of VTE, no increase in TG potential was shown in PRP or PPP, compared to healthy subjects [26]. The lack of association might be the result of the use of high concentrations of tissue factor (TF) in both PPP (30 picomolar, pM) and PRP conditions (6 pM), thus diminishing TG assay sensitivity to intrinsic plasma and/or platelet procoagulant features. In cancer patients at risk for VTE, platelet-dependent TG has been also of interest to explore underlying mechanisms predisposing to VTE. Increased platelet-dependent TG has been demonstrated in patients with myeloproliferative neoplasms, especially in carriers of the *JAK2V617F* mutation [27,28].

In addition to exploring platelet coagulant function in the pathophysiology of prothrombotic conditions, TG in PRP has also been proposed as a potential tool for monitoring antiplatelet therapy [4]. Altman showed that aspirin intake in healthy volunteers resulted in a prolonged lag time and time to peak and had no effect on thrombin peak level and ETP. The effect was more pronounced in subjects taking dual antiplatelet drugs, low-dose aspirin, and clopidogrel [29,30]. A study in stroke patients showed that the assessment of platelet-dependent TG is useful in measuring P2Y_12_ inhibition by clopidogrel therapy [31]. The effect of dual antiplatelet regimen, administered in patients with coronary artery disease following percutaneous coronary intervention, on platelet-dependent TG was also reported [32]. Interestingly, aspirin alone did not affect TG in PRP in a large cross-sectional population-based study [9]. In patients with left ventricular assist devices, the addition of aspirin to the anticoagulant regimen was found to minimally modulate platelet-dependent TG [33]. In an experimental model with in vitro addition of platelet inhibitors to PRP, aspirin and dipyridamole showed no effect on platelet-dependent endogenous thrombin potential [34]. On the other hand, studies have shown effects of other antithrombotic agents affecting mechanisms beyond cyclooxygenase (COX)-2 inhibition, such as protein disulfide isomerase inhibitors which reduce platelet-dependent thrombin generation by blocking the generation of platelet FVa [35,36]. Recent work explored the use of collagen-induced platelet aggregation (agPRP) for TG measurements in the presence of FXa inhibitors. Compared to resting PRP, agPRP was shown to enable full platelet potential to support TG in an anticoagulated setting [37]. These findings highlight the distinction between platelet aggregation and procoagulant functions, and support the use of platelet-dependent TG as a tool to investigate drug effects on a cell (platelet)-based model of coagulation.

## 3. Platelet-Dependent Thrombin Generation in Bleeding Diathesis

Patients with a defect in platelet function or coagulation, which can be congenital or acquired, have an increased risk of bleeding as a consequence of impaired clot formation. An accurate assessment of the hemostatic potential of blood is important for diagnosing and predicting bleeding risk and monitoring treatment with hemostatic agents. Like for prothrombotic conditions, there is a need for an integrative test that allows evaluation of the interplay between the different components of the hemostatic system. So far, limited evidence is available on the applicability of the TG test, performed in the presence of platelets, in bleeding disorders. Most of the studies that are available report on TG in inherited bleeding disorders, such as hemophilia, von Willebrand disease (VWD), and other coagulation factor deficiencies.

In hemophilia A and B, the impairment in TG is likely due to the absence of the FVIII/FIX tenase complex that mediates FX activation on the platelet surface. It is known that coagulation factor levels, TG parameters, and the clinical bleeding phenotype are not always in alignment. In patients with hemophilia A, a variation in TG was observed in patients with comparable levels of FVIII coagulant activity (FVIII:C) and the presence of platelets enhanced this variation when FVIII:C levels were less than 1–5% [38]. Among severe hemophiliacs, there is heterogeneity in the clinical phenotype, where a mild bleeding tendency was shown to be associated with higher ETP values in PRP [39]. Siegemund et al. demonstrated that TG (ETP) in PRP from patients with severe hemophilia A was not significantly different from patients with severe hemophilia B (FVIII/FIX:C < 2%), although the influence of platelet count on TG parameters appeared to be stronger in hemophilia B. Typically, the effect of platelets on ETP becomes less pronounced with higher FVIII or FIX levels [40].

In a study of 53 patients with different types of von Willebrand disease (VWD), TG in PRP was substantially decreased and delayed compared to healthy controls and was strongly dependent on the FVIII:C level. VWD patients with low thrombin peak values had a significantly higher risk of bleeding [41]. Evidence is inconclusive on the additional role of VWF in TG when FVIII:C levels are normal [41,42]. In another study, TG in the presence of platelets was only slightly reduced in patients with type-3 VWD (*n* = 9) receiving regular prophylaxis compared to patients who did not receive prophylaxis, indicating that platelets can partly compensate for the defect [43].

The fact that the inclusion of platelets in the TG assay is important for optimal evaluation of the bleeding tendency in patients is confirmed by studies in patients with severe congenital FV deficiency associated with hemorrhagic diathesis of varying degrees. Patients with undetectable plasma FV may contain functional FV in their platelets, which in combination with a low tissue factor pathway inhibitor (TFPI) level can still support sufficient TG [44]. This suggests that the amount of residual FV in platelets is a determinant for the severity of the bleeding phenotype [45].

Results on TG in PRP in patients with FXI deficiency demonstrated the ability of the assay to distinguish severe bleeding from non-bleeding individuals, which was independent of the FXI activity levels. In patients with a higher bleeding risk, TG was significantly lower (thrombin peak) and delayed (lag time, velocity) [46]. On the contrary, in a prospective study in 39 patients with heterozygous FXI deficiency, TG was not different between bleeding and non-bleeding patients [47]. The suggestion that the contradictory findings might be due to differences in assay/sample conditions is strengthened by the work of Pike et al., who showed that optimal differentiation in bleeding phenotype in FXI deficiency can be achieved by measuring TG in the presence of platelets when contact activation is inhibited [48].

Thrombin generation is not only applied to estimate the bleeding risk in patients with hemostatic defects, but also to monitor replacement therapy that is aimed at preventing or treating bleeding episodes. FVIII infusion in patients with hemophilia A increased the initial rate of TG in PRP, although in this study not all TG parameters could be measured because of severe impairment when FVIII levels declined over time [49]. Glanzmann thrombasthenia, characterized by a deficiency or dysfunction of the platelet integrin α_IIb_β_3_ leading to defective platelet aggregation, is associated with a bleeding phenotype ranging from mild to severe. All parameters of TG were decreased in patients with Glanzmann disease, but the test appeared insensitive to in vitro supplementation with fibrinogen and FXIII [50]. A recent study in 24 patients with Glanzmann disease provided evidence for significant improvement in TG in PRP with low concentrations of spiked recombinant factor VIIa (rFVIIa), which indicated that the TG test might be suitable to tailor rFVIIa therapy in these patients [51]. These findings are consistent with the reported improvement in TG after in vitro addition of prothrombin complex concentrate under conditions of hemodilution, whereas the addition of fibrinogen concentrate had no additional effect [52]. In line with this, TG (thrombin peak, ETP) was significantly enhanced in patients with dilutional coagulopathy after transfusion with fresh frozen plasma. Interestingly, thrombin peak levels were lower in patients who experienced bleeding during or after surgery compared to non-bleeding patients [53]. In another study on postoperative blood loss, TG in PRP was also found to predict bleeding, even when performed intraoperatively after heparin administration [54].

Altogether, these studies support the use of TG in the prediction of individual bleeding risk and the management of treatment in both congenital and acquired impairments of hemostatic function; however, optimization of assay conditions (TF concentration, inhibition of contact activation, etc.) is essential to obtain optimal sensitivity of the TG assay for specific hemostatic defects.

## 4. Uncertainties and Limitations of Thrombin Generation in the Presence of Platelets

Whereas the TG assay has been a method available since the early 1950s, the investigation of TG in PRP was made widely applicable and sparked a significant interest only after the introduction of the fluorogenic automated assay [4]. Despite the available evidence we summarized, there is still less information on the practical application and clinical relevance of the TG assay, mainly due to lack of prospective data linking TG parameters in PRP with clinical events, both thrombosis and bleeding. The few prospective studies available are often hampered by low patient numbers, meaning that the clinical predictive value of the TG assay still remains uncertain.

Performing TG in PRP has been challenging for several reasons. Unlike samples without platelets, TG in PRP requires immediate investigation. To increase practical applicability, stored PRP has been a subject of investigation. Whereas TG results in frozen–thawed PRP were significantly higher compared to fresh PRP obtained from the same healthy subjects, several features including the detection of activated protein C resistance were comparable for the two sample conditions [55]. Recent work including healthy subjects and individuals at risk for bleeding showed poor agreement between TG results from fresh and frozen–thawed PRP [56]. Whether TG variations depend on individual functional characteristics of platelets that differ with the underlying pathophysiological conditions is not yet understood.

Pre-analytical conditions greatly influence TG parameters and standardization of TG in PRP is necessary to enable a wide clinical use of the assay [57]. Data from animal and human studies clearly showed that platelet count in the PRP sample significantly impacts TG parameters [58,59]. A consensus that platelet counts between 150 × 10^9^/L to 400 × 10^9^/L results in no significant differences of TG parameters has been established and a platelet count of 150 × 10^9^/L has been shown to give stable results in terms of ETP in PRP. Extracellular vesicles in plasma (PPP and PRP), present in higher amounts in pathological conditions associated with cellular activation, can also influence TG by increasing the available procoagulant surface and activating the extrinsic or intrinsic coagulation pathway [60]. Exogenous addition of TF is another potential factor of variation between TG results. Low TF or absence of TF was linked to higher variation of TG results in PRP, while increasing TF concentrations showed no effect on the amount of TG formed [58]. Contact activation has been additionally shown to affect platelet-dependent TG, particularly when very low exogenous TF was used. The addition of corn trypsin inhibitor was associated with improved assay variability and enabled precise sample measurements up to 6 h after blood sampling in a small sample of healthy individuals [61]. These aspects make it difficult to compare TG results in PRP from different studies. Indeed, the majority of clinical studies discussed in this article have followed the recommendations in terms of platelet concentration in the PRP samples, by the use of adjusted PRP to 150,000/µL platelets. The use of a trigger was not uniform, varying from studies using TF at different concentrations and from different companies to studies using triggers other than TF, such as α-thrombin at low concentrations, to activate platelets in PRP. As the measurements in PRP require fresh samples, it is expected that prompt TG measurements were performed; however, very few studies reported the time of plasma preparation and initiation of TG measurements.

Mouse models have become increasingly important in thrombosis and hemostasis research and TG as a global coagulation assay has been frequently applied by the scientists. Mouse blood samples differ significantly from human samples, demanding particular adaptations of the TG assay [59]. Despite the obvious discrepancies, translation to humans remains important for elucidating specific mechanisms of platelet–thrombin interaction.

Further efforts on standardization are required that would enable merging results from different studies and thus improving the translation and clinical interpretation of TG in the presence of platelets. The use of standardized reagents and a highly automated process integrating different steps of TG measurement, including PRP separation, platelet count adjustment, and dispensing of plasma and reagents would highly contribute to uniformity of results across different laboratory settings and clinical conditions. A recently developed, fully automated TG analyzer (ST-Genesia, Stago, Asnières-sur-Seine, France), when tested using PFP with strict control of temperature, volumes, and normalization of each parameter against a reference plasma, offered increased reproducibility of the TG measurements [62,63]. Further investigations are needed to determine if the same principle could be applied to platelet-dependent TG.

## 5. Conclusions

Fluorogenic TG measurement, an automated and easy-to-use technology, has proven attractive to assess the interactions between platelets and coagulation factors. To reach a role as a potential biomarker in disease development or progression, large prospective multicenter studies with clearly defined clinical endpoints are required. Despite several studies addressing pre-analytical and analytical sources of variations, TG in the presence of platelets still lacks standardization. Even though the results of existing studies implicated an important and probably underappreciated role of platelet-dependent thrombin formation, its precise role in clinical practice remains inconclusive. With the important recognition for the improved net clinical benefit of the combined use of antiplatelet and anticoagulant agents in cardiovascular prevention, studying the effects of dual pathway inhibition on platelets by the TG assay also becomes an interesting issue that could potentially contribute to a type of personalized medicine. Presently, the platelet-dependent TG assay remains as an important research tool to investigate individual functional characteristics of platelets in relation to various clinical conditions, for both clinicians and researchers in clinical and experimental studies.

## Figures and Tables

**Figure 1 jcm-09-00092-f001:**
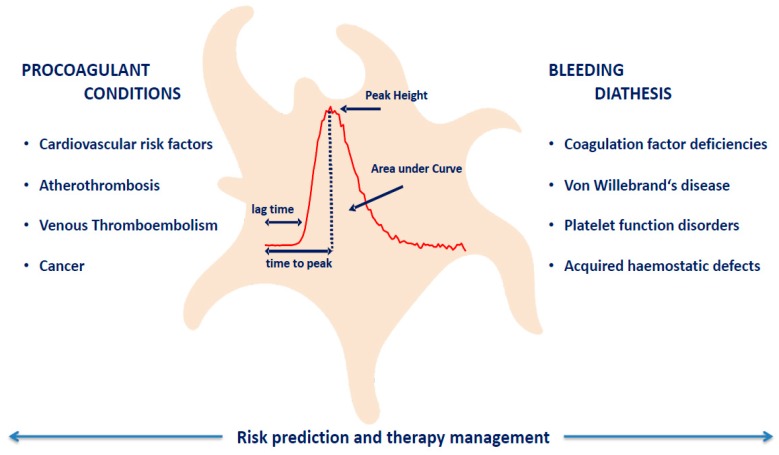
Thrombin Generation: Clinical Applications.

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
