# Peer review of "Clinical Applications, Pitfalls, and Uncertainties of Thrombin Generation in the Presence of Platelets"

_jcm, 2019, doi:10.3390/jcm9010092_

Round 1

Reviewer 1 Report

The authors have provided an interesting review of the clinical applications of the thrombin generation assay.  This work examines the usefulness of this test in assessing the diagnosis and/or treatment of a variety of hyper- and hypocoagulable conditions as well as in the monitoring of antithrombotic therapies. 

For this reason it will likely be of interest to your readership and would be suitable for publication in the journal of clinical medicine after some minor revisions.

Whilst the work covers the clinical application well, I believe it could be made more accessible to a wider audience by having a short paragraph (possibly with a figure) explaining how the assay is performed. 

Currently there is some indication of this throughout the text (e.g. fluorogenic assay, automated) but I think there needs to be a short section early on explaining this more explicitly. 

On a more minor note, some text editing is required to remove some awkardly-phrased sentences (e.g. L160 ....in plasma of nine type 3 VWD patients receiving either or not regular prophylaxis..).

Author Response

The authors have provided an interesting review of the clinical applications of the thrombin generation assay.  This work examines the usefulness of this test in assessing the diagnosis and/or treatment of a variety of hyper- and hypocoagulable conditions as well as in the monitoring of antithrombotic therapies.

For this reason it will likely be of interest to your readership and would be suitable for publication in the journal of clinical medicine after some minor revisions.

Whilst the work covers the clinical application well, I believe it could be made more accessible to a wider audience by having a short paragraph (possibly with a figure) explaining how the assay is performed. Currently there is some indication of this throughout the text (e.g. fluorogenic assay, automated) but I think there needs to be a short section early on explaining this more explicitly.

Authors: Thank you for this important comment. We agree that explicit introduction of the most frequently used TG assay, is missing in our manuscript. We have updated the manuscript and included a short paragraph explaining the assay principle on page 2, as follows: “The calibrated automated thrombogram (Thrombinoscope BV, Maastricht, The Netherlands), the most frequently used TG assay, employs a standardized testing procedure where readings from the fluorometer are recorded and calculated by means of Thrombinoscope software (Thrombinoscope BV). In principle, the measurements in PRP are assessed in fresh material whereas TG in PFP is assessed in frozen, stored material. For TG measurements in PRP, coagulation is triggered by adding 20μl exogenous tissue factor (TF) to 80μl PRP (with adjusted platelet concentration of 150,000 platelets/μl), whereas for the measurements in PFP, 20μl exogenous TF together with 4μM phospholipids are added to 80μl plasma. After 10 minutes prewarming at 37ºC in the fluorimeter, the reaction is started by adding 20μl of a low-affinity fluorogenic substrate for thrombin (Z-Gly-Gly-Arg-AMC) and calcium chloride mixture (FluCa). TG is measured as a function of a calibration signal obtained in a sample from the same plasma (PRP or PFP) after addition of a fixed amount of thrombin–α2-macroglobulin complex (20μL of Thrombin Calibrator) and 20μL of FluCa. The TG method is extensively reviewed in [5-7].”

On a more minor note, some text editing is required to remove some awkwardly-phrased sentences (e.g. L160 ....in plasma of nine type 3 VWD patients receiving either or not regular prophylaxis...).

Authors: Thank you for this remark. We have adjusted the sentence and edited the text on page 7 as follows: “In another study, TG in the presence of platelets was only slightly reduced in type 3 VWD patients (n=9) receiving either or not regular prophylaxis, indicating that platelets can partly compensate for the defect [40].”

Reviewer 2 Report

In the present review article titled "Clinical applications, pitfalls, uncertainties of thrombin generation in presence of platelets" the authors summarize current available clinical data on thrombin generation, its application and the correlation of TG data with bleeding phenotypes as well as thrombotic complications.

Assay systems for a combined analysis of platelets and the coagulation system are still limited, TG measurement representing one of the few established methods. Recent data point more and more to an important role of platelets in thrombin generation under prothrombotic conditions as well as under limited coagulatory/hemostatic function.

However, assay protocols are still highly variable, and a standardization of the protocols is strongly required to allow better comparison of the obtained results.

The review focuses on the detection of bleeding phenotypes as well as prothrombotic conditions by TG and transferability of TG results to the clinics.

The review is written in a concise way and follows a logical progression.

Can a role of MP in PPP studies be excluded? Please discuss in the review. How comparable are the different CAT protocols of studies analysed in this review? E.g. with regard to used platelet concentration, TF concentration, time after blood withdrawal? Recently, an in vitro study on the effects of Rivaroxaban on TG (Makhoul et al., JCM 2019) was published.

Could you please include this as well in the review? How transferable are results from animal - mostly mouse - studies to the human system/clinical situation? Especially with regard to different assay protocols used for human and mouse system.

Differences in the TG assay protocol are huge. Is TG potent enough to identify bleeders from non bleeders in a stand alone measurement? Is there any advice on positive/negative controls?

The review focuses on the transferability of TG data to the clinical situation (prothrombotic, bleeding conditions). Which factors are mostly altered (ETP, Thrombin peak) and what conclusions can be drawn from this?

How could the standardization of TG measurements be improved? Can you give any advice?

Author Response

In the present review article titled "Clinical applications, pitfalls, uncertainties of thrombin generation in presence of platelets" the authors summarize current available clinical data on thrombin generation, its application and the correlation of TG data with bleeding phenotypes as well as thrombotic complications. Assay systems for a combined analysis of platelets and the coagulation system are still limited, TG measurement representing one of the few established methods. Recent data point more and more to an important role of platelets in thrombin generation under prothrombotic conditions as well as under limited coagulatory/hemostatic function. However, assay protocols are still highly variable, and a standardization of the protocols is strongly required to allow better comparison of the obtained results.

The review focuses on the detection of bleeding phenotypes as well as prothrombotic conditions by TG and transferability of TG results to the clinics.

The review is written in a concise way and follows a logical progression.

Can a role of MP in PPP studies be excluded? Please discuss in the review. How comparable are the different CAT protocols of studies analysed in this review? E.g. with regard to used platelet concentration, TF concentration, time after blood withdrawal? Recently, an in vitro study on the effects of Rivaroxaban on TG (Makhoul et al., JCM 2019) was published. Could you please include this as well in the review?

Authors: Thank you for these important points.

Regarding the first point, excluding the role of MP in PPP would require ultra-high centrifugation of the plasma samples (>30,000 g), higher than recommended speed (10,000 g) for platelet free plasma (PFP) extraction. In addition, it has been shown that at lower concentration of TF as a trigger for the reaction, MP contribution is of greater magnitude. In this review we focus on TG in PRP and we do not discuss results on TG in PPP. Nonetheless, in pathological conditions where there is abundant cellular activation, the number of MPs is increased, which can affect TG in PRP by increasing the available procoagulant surface and activating the extrinsic or intrinsic coagulation pathway. This is now stated in the text on page 9 as follows: “Extracellular vesicles in plasma (PPP and PRP), especially present in higher amounts in pathological conditions associated with cellular activation, can also influence TG by increasing the available procoagulant surface and activating the extrinsic or intrinsic coagulation pathway [59].”

Regarding the second point on how comparable are different CAT protocols applied in the clinical setting we have added the following lines on page 9, under paragraph on “Uncertainties and limitations of thrombin generation in presence of platelets” as follows: “These aspects make it difficult to compare TG results in PRP from different studies. Indeed, the majority of clinical studies discussed in this article have followed the recommendations in terms of platelet concentration in the PRP samples, by the use of adjusted PRP to 150,000/µl platelets. Differently, the use of a trigger was not uniform, varying from studies using TF at different concentrations and from different companies to studies using triggers other than TF as e.g. α-thrombin at low concentrations to activate platelets in PRP. As the measurements in PRP require fresh samples, it is expected that prompt TG measurements were performed. However, very few studies reported the time of plasma preparation and initiation of TG measurements.”

Regarding the third point, when preparing our manuscript, the paper by Makhoul et al. published recently at JCM was not yet available. We have now added this work on page 6, as follows: “Recent work explored the use of collagen-induced platelet aggregation (agPRP) for TG measurements in presence of Factor Xa inhibitors. Compared to resting PRP, agPRP was shown to enable full platelet potential to support TG in an anticoagulated setting [37].” 

How transferable are results from animal - mostly mouse - studies to the human system/clinical situation? Especially with regard to different assay protocols used for human and mouse system.

Authors: This is an important point. We have discussed how transferable results are on page 10, first paragraph as follows: “Mouse models have become increasingly important in thrombosis and hemostasis research and TG as a global coagulation assay has been frequently applied by the basic scientists. Mouse blood samples differ significantly from human samples, demanding particular adaptation of the TG assay applied in mouse blood compared to human blood samples [58]. Despite the obvious discrepancies, translation to humans remains important for elucidating specific mechanisms of platelet-thrombin interaction.”

Differences in the TG assay protocol are huge. Is TG potent enough to identify bleeders from non bleeders in a stand alone measurement? Is there any advice on positive/negative controls?

Authors: So far, the majority of studies linking TG parameters in PRP to bleeding events are based on retrospective data. The few prospective studies available are often hampered by low patient numbers, meaning that the clinical predictive value of the TG assay still remains uncertain. This is now stated in the text in the section ‘Uncertainties and limitations of thrombin generation in the presence of platelets’ on page 8-9. 

The review focuses on the transferability of TG data to the clinical situation (prothrombotic, bleeding conditions). Which factors are mostly altered (ETP, Thrombin peak) and what conclusions can be drawn from this?

Authors: Thank you for this comment. In our opinion, the value of TG assay is that it provides several parameters on thrombin formation with time-dependent variables as the lag time and time to peak and variables giving information on the amount of TG (peak height and ETP). In our opinion it is very valuable to report both kinetic- and amount-dependent variables, as both could be altered in different clinical scenarios.

How could the standardization of TG measurements be improved? Can you give any advice?

Authors: Thank you for this question. On page 10, we have added the following: “Use of standardized reagents, increased automated process integrating different steps of TG measurements including PRP separation, platelet count adjustment and dispensing of plasma and reagents would highly contribute to uniformity of results across different laboratory settings and clinical conditions.”
